# Sexual and reproductive health rights: A cross-sectional study of knowledge and practice among the married women of reproductive age residing in Besishahar Municipality, Nepal

Laxmi Gautam[1]*, Sirjana Adhikari[1], Amrit Bist[2], Sujan Gautam[1,3]

**1** Manmohan Memorial Institute of Health Sciences, Kathmandu, Nepal, **2** Patan Academy of Health Sciences, Lalitpur, Nepal, **3** Institute of Fundamental Research and Studies (InFeRS), Kathmandu, Nepal

* laxmi.dp26@gmail.com

## Abstract

Married women of reproductive age (MWRA) can experience violations of their sexual and reproductive health rights (SRHR). Adequate knowledge and understanding of SRRs are critical to their ability to protect themselves. This study assessed the knowledge and practice of SRHRs among the MWRA residing in Besishahar Municipality. A cross-sectional study was conducted among the 342 MWRA in Besishahar Municipality using a pre-tested structured questionnaire through face-to-face interview. The collected data were analyzed using SPSS IBM version 16. The logistic regression model was applied to examine the factors associated with the outcome variable using an adjusted odds ratio with a 95% CI, and a p-value <0.05 was considered statistically significant. Among the respondents, 47.7% had adequate knowledge on SRHR and 41.5% had good practice. Women married at age ≥ 18 years were 2.075 times more likely to have adequate knowledge of SRHR compared to women who had early marriages (<18 years) (aOR=2.075, 95% CI = 1.16-3.69) while women married at age ≥ 18 years were 1.82 times likely to had good practice than those married at age 18 years (aOR=1.82, 95% CI = 1.02-3.24). Respondents involved in formal sectors were 1.834 times more likely to have adequate knowledge of SRHR compared to informal sectors (aOR=1.834, 95% CI = 1.15-2.90) which was (aOR=1.518, 95% CI = 0.96-2.39) in case of good practice. The odds of having adequate knowledge was 2.51 among the respondents who were above the poverty line (aOR=2.511, 95%CI = 1.52-4.14). MWRA who had adequate knowledge of SRHR were 3.234 times more likely to have good practice of SRHR compared to women who had inadequate knowledge of SRHR (aOR=3.234, 95% CI = 1.85-6.56). A large proportion of married women of reproductive age did not have adequate knowledge about SRHR, and their practice was poor. So intervention focusing on the promotion of knowledge on SRHR is essential at the community level.

**Data availability statement:** All relevant data are within the paper and its Supporting Information files.

**Funding:** The authors received no specific funding for this work.

**Competing interests:** The authors have declared that no competing interests exist.

## Introduction

Sexual and reproductive health rights (SRHR) are fundamental components of the broader human right to the highest attainable standard of physical and mental health [1]. These rights apply universally to all individuals, irrespective of age, gender, sexual orientation, or other personal characteristics. SRHR encompasses the freedom to make informed choices about one's own sexuality and reproductive health, as long as these decisions do not infringe upon the rights of others. Various entitlements involve the scope of SRHR, which includes the right to life, liberty, and personal security; access to comprehensive health care and accurate information, protection against discrimination in the distribution and accessibility of health resources [2,3]. It also mentioned personal autonomy and privacy in making sexual and reproductive choices and ensuring that individuals should have the right to informed consent and confidentiality regarding their health-related information. The formal recognition of SRHR as basic human rights was made at the International Conference on Population and Development (ICPD) in Cairo in 1994, and this became a milestone for health and human rights advocates all over the world [4]. In particular, women's SRHR are an assertion of the right to be in control of the choices that women make over their reproductive lives; thus, the ability to exercise free and responsible choices regarding their sexual and reproductive health free from coercion, discrimination, or violence [5].

The Sexual and Reproductive Health (SRH) status in developing countries remains unsatisfactory and seriously impinges on the quality of life for women. This status is due to several interwoven factors involving unequal access to appropriate information and basic health and essential health services. The age at marriage on average was 17.2 years, representing a characteristic feature of early marriages that exposes these young girls to early pregnancy with associated health risks. The deeply ingrained cultural beliefs, dominant social norms, and restrictive gender roles in the society further constrain women's autonomy to make reproductive choices. Similarly, low literacy rates as high as 42.0% impede one's ability to make truly informed decisions about health and well-being [6]. The unmet need for family planning was 24.6%, which depicts a deficiency not only in access to contraception but also in reproductive health education [6]. This unmet need increases the risk of unwanted pregnancies, unsafe abortions, and maternal complications. Shockingly, up to 80.0% of unsafe abortions and deliveries are done by untrained personnel, posing serious risks to maternal health and contributing to avoidable deaths [6]. Sexual and reproductive health problems taken together contribute to about 20.0% of the global burden of disease among women, underlining their critical impact on women's overall health [6]. Full exercise of the SRHR may provide a panacea to some of these daunting challenges. These will significantly reduce maternal mortality and improve the economic and social status of women to achieve better gender equity. Knowledge and understanding of rights related to SRHR nurture fairness, reduce health disparities, and allow men and women to make rightful choices regarding their sexual and reproductive lives. This includes an understanding of male and female anatomy, the use of contraceptives, prevention of STIs and HIV/AIDS, and proper maternal care in

case of pregnancy and delivery. These very factors, when addressed, enable SRHR to improve not only individual health outcomes but also the broader landscape of sexual and reproductive health [7].

Nepal has made huge commitments in adopting major international frameworks, like the International Conference on Population and Development (ICPD) in 1994, the Beijing Platform for Action in 1995, and the Millennium Development Goals (MDGs) in 2000 [8]. All these commitments relate to recognizing that Nepal considers SRHR a core part of sustainable development and gender equality. Moreover, the Nepalese government ratified the Convention on the Elimination of All Forms of Discrimination Against Women (CEDAW) in 1991 with no reservations and the Convention on the Rights of the Child (CRC) in 1990, hence pledging a high level of political commitment toward full implementation and further strengthening the mandate to promote reproductive rights, equity in gender perspectives, and childcare [8]. The Constitution of Nepal guarantees reproductive rights at the national level in clear terms: "Every woman shall have the right to reproductive health and other reproductive matters." This constitutional provision has created an enabling environment whereby women can claim and exercise control over their own bodies and reproductive choices, thus laying a legal foundation for comprehensive reproductive health programs [8]. The National Women Commission, established under the National Commission for Women Act, 2074 BS, performs an important function in monitoring legislation on women's rights. Its other priority areas of work include assessing the country's compliance with international treaties and agreements, making recommendations to the government for effective policy enforcement [9]. The commitment of the Government of Nepal has further been solidified by the enacting of the Safe Motherhood and Reproductive Health Rights Act, 2018, which ensures access to safe motherhood services, reproductive health care, and protection of reproductive rights under law. Similarly, at the Nairobi Summit, Nepal declared its commitment to the full implementation of the ICPD Programme of Action. More recently, Nepal has also been pursuing the Sustainable Development Goals (SDGs), especially Goal 3 (Good Health and Well-being) and Goal 5 (Gender Equality), which address ensuring universal access to reproductive health services and undertaking all possible measures to eliminate discrimination based on gender [10]. Despite these rather progressive legal frameworks, gender-based discriminatory practices emanate from deep-seated traditions, a strongly engraved patriarchal societal system, and structural inequalities in Nepal. Many women still face various kinds of systemic obstacles that have challenged their capacities in exercising reproductive options. Their feelings of powerlessness and lowly status within a community exacerbate health concerns associated with access to maternal health care services, a high rate of gender-based violence, and unmet reproductive health education needs [11].

The burden of the ill reproductive health due to lack of awareness and underutilization of reproductive rights among the women is globally higher and also in Nepal, which makes women more vulnerable to ill health and maternal deaths. Providing knowledge on reproductive rights can prevent maternal deaths and improve women's status in society. In spite of all these commitments and efforts, the status of maternal health is not good in Nepal. The knowledge on SRHR and utilization of these among Nepalese women in Nepal were not well assessed in community settings. So, this study assessed the knowledge of sexual and reproductive health rights among reproductive-age women along with the utilization of those rights in the community. The study provides baseline insights about the status of knowledge and practices in SRHRs and their contributing factors among the women in developing countries like Nepal. It might help in addressing barriers to the utilization of sexual and reproductive health rights and be helpful to design and implement interventions to increase knowledge about SRHRs among Nepalese women, which can contribute to improving their sexual and reproductive health and can help attain SDG goals.

## Materials and methods

### Study design, population, and setting

A community-based cross-sectional study was conducted to assess the knowledge and practice about sexual and reproductive health rights among married women of reproductive age in the Besishahar Municipality, Lamjung district, Gandaki Province, Nepal. The Besishahar Municipality consists of 11,038 households with a total population of 38,232 residents, out of which 65.6% were married women with a median age of marriage of 18–20 years [12]. This municipality was

chosen because it reflects a wide range of socio-cultural and economic landscapes, representing most of the castes, ethnicities, and major religions of the nation. The female literacy rate was much lower than that of males, 74.6% compared to 88.8%, reflecting existing disparities influenced by economic inequality, educational gaps, and cultural norms [12]. These disparities reveal the importance of assessing how such factors influence the utilization of sexual and reproductive health rights among married women in this setting. This study was based on Strengthening the Reporting of Observational Studies in Epidemiology (STROBE) guidelines for reporting observational studies. We used the STROBE checklist as a framework to ensure comprehensive and standardized reporting of all relevant sections, including the study design, setting, participant selection, variable definitions, data sources, statistical methods, and interpretation of results.

## Sample size and sampling procedure

The sample size was calculated using Cochran's formula to estimate a proportion ($n = z^2pq/d^2$) at a 95% confidence interval and a 5% margin of error. A previous study in Kapan, Nepal, reported that 68.3% of the study population had knowledge on sexual and reproductive health rights [6]. Considering these proportions, the initial sample size was estimated to be 333, optimized to 366, adjusting for the 10% non-response rate. However, during the data collection, 10 participants did not provide informed consent, 7 were found sick on the day of data collection, and 7 study participants were not found in their home even in the follow-up visit. So, 24 participants were excluded from the study, resulting in a final sample of 342 married women of reproductive age.

All 11 wards of Besishahar Municipality were included, with the number of participants to be selected from each ward in proportion to the total number of married women aged 18–49 years in that ward according to data obtained from the municipality. A line listing of the households with eligible married women was prepared, from which participants were randomly selected using the 'RANDBETWEEN' function in Microsoft Excel. Only married women of reproductive age who gave informed consent were selected for the study, whereas those who did not provide informed consent, who were sick, and not found in their home in consecutive visits were excluded from the study. (S1 Text)

## Data collection process

Data collection was done through face-to-face interviews using a structured questionnaire from September 2023 to November 2023. Interviews were conducted in participants' households after obtaining informed consent, ensuring that participants were well-oriented about the objectives of the research study, their rights to withdraw at any time, and the voluntary nature of their participation. Interviews were conducted in a private area within the home, away from other household members, to maintain privacy and confidentiality. Assurance of confidentiality of responses and that they would only be used for research purposes was given. The interviews took approximately 35–40 minutes.

The data collector was a well-trained public health professional with experience in SRH. The tool was developed based on an extensive review of the English literature and then was translated into Nepali through a forward-backward method in consultation with subject experts in order to achieve content validity. To further ensure reliability, the tool was sent to an expert in reproductive health for review prior to data collection. A pre-test was done among 10% of similar populations in the Marshyandi Rural Municipality of Lamjung district that refined the clarity and precision of the question to ensure face validity. The reliability of the tool was assessed using Cronbach's alpha, yielding a value of 0.78. (S2 Text)

## Study variables

**Exploratory variables.**  The exploratory variables included socio-demographic and economic characteristics that could affect the knowledge and practice of sexual and reproductive health rights among the married women of reproductive age. These variables included the age of the participants that could affect their level of awareness and experience in terms of reproductive health and ethnicity due to diverse socio-cultural backgrounds within which health behaviors and access to information could be based. The religion was included for assessment as it often shapes beliefs, practices,

and attitudes about reproductive health. The type of family that a person had nuclear or joint and the type of marriage, arranged or love, were considered while assessing the family dynamics influencing decisions on reproductive health, along with the age at the time of first marriage. Educational variables included the participant's level of education. The occupation status of the respondent and her spouse was inquired into in order to get an understanding of how occupation and economic involvement may affect access to reproductive health facilities and decision-making at home. The economic status variable was included as a key one to assess the role of financial stability in either facilitating or hindering access to sexual and reproductive health services [13] Average income less than $1.90 per day is defined as below the poverty line and equal or more than $1.90 per day is defined as above the poverty line in reference to the World Bank [14].

**Outcome variables.** The outcome variables for this study were knowledge and practice on SRHR. The variables used to assess the level of knowledge on SRHR included information about meaning of sexual and reproductive health rights, women have right to obtain education, information, counseling and service relating to sexual and reproductive health, knowledge on right of women to have nutritious diet during pregnancy and Childbirth, knowledge on right of women to have physical rest during condition of pregnancy and childbirth, knowledge on right to be prevented and treated the reproductive health morbidity, knowledge on right to information and services of family planning, Knowledge on the right of protection from sexual abuse, reproductive harms and sexual discrimination, Knowledge on violation of sexual and reproductive health rights is punishable by law, and so on [15]. Each correct response was awarded 1 point, while incorrect or "don't know" responses received 0 points. The normality test was conducted using Kolmogorov Smirnov test and found that knowledge score does not follow normal distribution. As a result, the median was then calculated, which was 18. To differentiate the level as adequate and inadequate knowledge, more than the median was considered as adequate knowledge (≥18) and less than 18 was considered as inadequate knowledge.

Similarly, respondent's level on the practice of the SRHR score was calculated out of the practice-specific questions. The variables used to measure the practice included decision maker for marriage and childbirth, utilization of maternal health services, family planning methods and abortion services, involvement in health-related activities, and decision makers in all these activities [16–19]. Each correct response was awarded 1 point, while incorrect or "don't know" responses received 0 points. The median value was calculated as the data doesn't follow normal distribution and which was found to be 11. Good practice of SRHR was given to those respondents who scored greater than or equal to the median value (≥11) of the sum of correct response. Poor practice of SRHR was given to those respondents who scored below the median value (<11) of the sum of correct responses.

## Statistical analysis

The data collected were initially entered into Microsoft Excel, with a random 10% of the entered data manually reviewed to ensure accuracy, during which no data entry errors were identified. Following data verification, the data set was imported into the Statistical Package for the Social Sciences (SPSS) version 16 for statistical analysis. Descriptive statistics frequency and percentage were calculated for the background independent variables. The median and interquartile range were calculated for the knowledge and practice on SRHR. The association of different independent variables with the dependent variable was determined by cross-tabulation and the chi-square tests. The variables found to be statistically significant in chi-square test were included in the multivariate binary logistic regression for multivariable analysis to calculate the adjusted odds ratio (aOR). The variance inflation factor (VIF) test was performed among selected independent variables to manage the issue of multicollinearity. The VIF greater than two was taken as an indication of multicollinearity. The p-values <0.05 were taken as statistically significant. (S1 Data)

## Ethical statement

Ethical approval was taken from the Institutional Review Committee (IRC) of Manmohan Memorial Institute of Health Sciences (MMIHS-IRC 78/33). Permission was taken from the municipality, and written informed consent was taken

from each participant before data collection after explaining the objective, methods, and anticipated benefits of the study. Participants were informed about privacy and confidentiality of information as well as the right to withdraw and the right to refuse if they are interested to continue or not interested to answer, and there was no harm in doing so.

## Results

The median age of the respondents was 34 years, with majority of participants (51.46%) in age group 18–33 years followed by participants in age group 34–49 years. Respondents who belonged to Janajati and Chhetri each covered almost one-third of the respondents (32%), whereas more than two-thirds (70.2%) followed Hindu religion. Half of the respondents (52%) belonged to a nuclear family type, and more than two-thirds (69.0%) got arranged marriages. About one-fourth of the respondents (24.30%) had their first marriage before the age of 18 years. More than one-fourth (27.8%) of the respondents had completed secondary education, whereas nearly one-third (31.3%) of the respondents' husbands had completed higher education. Less than half (43.6%) of respondents were homemakers whereas their husbands' main occupations were private sector jobs (23.1%), and foreign employment (22.8%). Nearly two-thirds of the respondents (63.5%) were living above the poverty line (average family income ≥1.90$ per day) (Table 1).

The median knowledge score was 18 with an interquartile range of 6, where 47.7% of respondents had adequate knowledge (≥18) and 52.3% had inadequate knowledge (<18). In terms of SRHR practice, the median score was 11 with an Interquartile range of 2, with 41.5% of respondents exhibiting good practice (≥11) and 58.5% demonstrating poor practice (<11) (Table 2).

The bi-variate analysis found the significant association between age and knowledge of SRHR (p-value = 0.004). Similarly, the respondent's age at first marriage (p-value <0.001) was also found to be significantly associated with knowledge of SRHR. Occupational status of respondents showed a significant association with knowledge of SRHR (p-value = 0.001). Similarly, husband occupational status (p-value = 0.035), and education of respondent (p < 0.001) and education level of husband (p-value<0.001) were found to be significantly associated with the knowledge of SRHR. Economic status also showed a significant association with knowledge of SRHR (p-value<0.001). Discussion of SRH issues with others was found significantly associated with knowledge of SRHR (p-value = 0.001). The source of information from awareness programs/trainings also showed a significant association with knowledge of SRHR (p-value = 0.004). Other socio-demographic variables like ethnicity, religion, family type, and type of marriage showed no significant association with knowledge of SRHR (Table 3).

For multivariable analysis, the VIF test was done among the independent variables that were found to have a statistically significant association with knowledge on SRHR. The highest reported VIF was 1.639, indicating that there was no issue of multicollinearity. It was observed that married women of reproductive age 18–33 years, were 1.811 times more likely to have adequate knowledge of SRHR compared to women in the 34–49 age group (aOR=1.811, 95% CI = 1.145-2.864) adjusting for all other variables. Women who hadn't done early marriage were 2.075 times more likely to have adequate knowledge of SRHR compared to women who had early marriage (aOR=2.075, 95% CI = 1.162-3.691). Respondents who were involved in formal sectors were 1.834 times more likely to have adequate knowledge of SRHR compared to those who were involved in informal sectors (aOR=1.834, 95% CI = 1.158-2.904). Married women who belonged above the poverty line were 2.511 times more likely to have adequate knowledge compared to women belonged below the poverty line (aOR=2.511, 95% CI = 1.523-4.141) after adjusting the other variables. The respondents with a higher level of education were 7.367 (aOR=7.367, 95% CI = 2.34-13.4) times more likely to have adequate knowledge than the illiterate women. Similarly, the married women whose husbands had higher level of education were 1.872 (aOR=1.872, 95% CI = 1.20-4.87 times likely to have adequate knowledge as compared to the women whose husbands were illiterate. The respondents who received the information from the awareness program and training were 2.134 (aOR= 2.134, 95% CI = 1.72-5.67) times more likely to have adequate knowledge compared to those who did not receive information. (Table 4).

The study found that certain factors were significantly associated with the practice of Sexual and Reproductive Health and Rights (SRHR) using the chi-square test. Specifically, the type of marriage (p-value = 0.009), age at first marriage

**Table 1. Sociodemographic information of the respondents.**

| Characteristics | Frequency (n = 342) | Percentage (%) |
|---|---|---|
| **Age (**Median ± IQR = 34 ± 6 years) | | |
| 18-33 | 176 | 51.46 |
| 34-49 | 166 | 48.54 |
| **Ethnicity** | | |
| Brahmin | 55 | 16.10 |
| Chhetri | 110 | 32.20 |
| Janajati | 111 | 32.50 |
| Dalit | 45 | 13.20 |
| Others | 21 | 6.00 |
| **Religion** | | |
| Hindu | 240 | 70.20 |
| Buddhist | 59 | 17.30 |
| Christian | 24 | 7.00 |
| Muslim | 5 | 1.50 |
| Others | 14 | 4.10 |
| **Family type** | | |
| Nuclear | 178 | 52.00 |
| Joint | 136 | 39.80 |
| Extended | 28 | 8.20 |
| **Marriage type** | | |
| Love marriage | 106 | 31.00 |
| Arrange marriage | 236 | 69.00 |
| **Age at first marriage** | | |
| < 18 years | 83 | 24.30 |
| ≥18 years | 259 | 75.70 |
| **Educational level of respondents** | | |
| Illiterate (cannot read and write) | 57 | 16.70 |
| Literate only (can read and write but no formal education) | 69 | 20.20 |
| Primary education | 88 | 25.70 |
| Secondary education | 95 | 27.80 |
| Higher education | 33 | 9.60 |
| **Educational level of husband** | | |
| Illiterate (cannot read and write) | 7 | 2.00 |
| Literate only (can read and write but no formal education) | 23 | 6.70 |
| Primary education | 70 | 2.50 |
| Secondary education | 135 | 39.50 |
| Higher education | 107 | 31.30 |
| **Occupational status of respondents** | | |
| Unemployed | 6 | 1.80 |
| Services | 20 | 5.80 |
| Business | 57 | 16.70 |
| Homemaker | 149 | 43.60 |
| Farmer | 27 | 7.90 |
| Private sectors | 53 | 15.50 |
| Health sectors | 16 | 4.70 |
| Others | 14 | 4.10 |

*(Continued)*

**Table 1.** (Continued)

| Characteristics | Frequency (n = 342) | Percentage (%) |
|---|---|---|
| **Occupational status of husband** | | |
| Unemployed | 27 | 7.90 |
| Government job | 66 | 19.30 |
| Business | 36 | 10.50 |
| Farmer | 28 | 8.20 |
| Private sector job | 79 | 23.10 |
| Foreign employment | 78 | 22.80 |
| Health sectors | 12 | 3.50 |
| Others | 16 | 4.70 |
| **Economic status** | | |
| Below the poverty line | 125 | 36.50 |
| Above the poverty line | 217 | 63.50 |

**Table 2.** Respondent's level of knowledge and practice of SRHR.

| Variables | Frequency (n = 342) | Percentages (%) |
|---|---|---|
| **Level of Knowledge of SRHR** | | |
| Median ± IQR = 18 ± 6 | | |
| Adequate (≥18) | 163 | 47.7% |
| Inadequate (<18) | 179 | 52.3% |
| **Practice of SRHR** | | |
| Median ± IQR = 11 ± 2 | | |
| Good (≥11) | 142 | 41.50 |
| Poor (<11) | 200 | 58.50 |

(p-value = 0.001), occupational status of respondents (p-value = 0.012), economic status (p-value <0.001), education level of husband (p-value = 0.006) were significantly linked to SRHR practices. Participation at health clubs/mothers groups showed a significant association with the practice of SRHR (p-value<0.001). Similarly, discussion of SRH issues with others was found significantly associated with practice of SRHR (p-value = 0.001). There was a significant association between the source of information from awareness programs/trainings and the practice of SRHR (p-value = 0.032). However, age, ethnicity, religion, family type, education level of respondents, and husband's occupational status showed no significant association with SRHR practices (Table 5).

The variables that were found significant with the practice of SRHR using the chi-square test were subjected to multivariate analysis after the VIF test. Women who had done arranged marriage were 1.768 times more likely to have good practice of SRHR compared to women who had love marriage (aOR=1.768, 95% CI = 1.073-2.914) adjusting for the other variables. Respondents who hadn't done early marriage were 1.821 times more likely to have good practice compared to respondents who had early marriage (aOR=1.821, 95% CI = 1.023-3.244). Women who belonged above the poverty line were 1.900 times more likely to have good practice compared to women who belonged below the poverty line (aOR=1.900, 95% CI = 1.155-3.125) adjusting all other variables in a multiple logistic regression model. The women who participated in the health club/mothers group were 4.213 times (aOR= 4.213, 95% CI = 2.35-7.69) more likely to practice SRHR than those who did not participate. The married women who discuss SRH issues with others were 3.206

**Table 3. Bivariate analysis of independent variables with knowledge of SRHR.**

| Variables | Knowledge of SRHR | | χ 2 | p-value |
|---|---|---|---|---|
| | Adequate n (%) | Inadequate n (%) | | |
| **Age** | | | | |
| 18-33 | 97 (55.1) | 79 (44.9) | 8.074 | 0.004* |
| 34-49 | 66 (39.8) | 100 (60.2) | | |
| **Ethnicity** | | | | |
| Brahmin/Chhetri | 79 (47.9) | 86 (52.1) | 2.749 | 0.253 |
| Janajati | 58 (52.3) | 53 (47.7) | | |
| Others | 26 (39.4) | 40 (60.6) | | |
| **Religion** | | | | |
| Hindu | 41 (42.3) | 56 (57.7) | 1.579 | 0.205 |
| Non-Hindu | 122 (49.8) | 123 (50.2) | | |
| **Family type** | | | | |
| Nuclear family | 76 (46.3) | 88 (53.7) | 0.220 | 0.639 |
| Others | 87 (48.9) | 91 (51.1) | | |
| **Type of marriage** | | | | |
| Love marriage | 49 (46.2) | 57 (53.8) | 0.127 | 0.722 |
| Arrange marriage | 114 (48.3) | 122 (51.7) | | |
| **Educational level of respondent** | | | | |
| Illiterate | 32 (56.1) | 25 (43.9) | 36.64 | <0.001* |
| Literate only | 32 (46.4) | 37 (53.6) | | |
| Primary education | 23 (26.1) | 65 (73.9) | | |
| Secondary education | 48 (50.5) | 47 (49.5) | | |
| Higher education | 28 (84.8) | 5 (15.2) | | |
| **Husband's educational level** | | | | |
| Illiterate | 1 (14.3) | 6 (85.7) | 38.86 | <0.001** |
| Literate only | 5 (21.7) | 18 (78.3) | | |
| Primary education | 18 (26.1) | 51(73.9) | | |
| Secondary education | 67 (49.3) | 69 (50.7) | | |
| Higher education | 72 (67.3) | 35 (32.7) | | |
| **Age at first marriage** | | | | |
| < 18 years | 23 (27.7) | 60 (72.3) | 17.486 | <0.001* |
| ≥18 years | 140 (54.1) | 119 (45.9) | | |
| **Occupational status** | | | | |
| Formal sectors | 105 (56.1) | 82 (43.9) | 11.919 | 0.001* |
| Informal sectors | 58 (37.4) | 97 (62.6) | | |
| **Husband's occupational status** | | | | |
| Formal sectors | 155 (49.4) | 159 (50.6) | 4.455 | 0.035 |
| Informal sectors | 8 (28.6) | 20 (71.4) | | |
| **Economic status** | | | | |
| Below the poverty line | 38 (30.4) | 87 (69.6) | 23.529 | <0.001* |
| Above the poverty line | 125 (57.6) | 92 (42.4) | | |
| **Discussion of SRH issues with others** | | | | |
| Yes | 141 (43.8) | 181 (56.2) | 11.668 | 0.001* |
| No | 1 (5) | 19 (95) | | |

*(Continued)*

**Table 3.** (Continued)

| Variables | Knowledge of SRHR | | χ 2 | p-value |
|---|---|---|---|---|
| | Adequate<br>n (%) | Inadequate<br>n (%) | | |
| **Source of information from awareness programs and trainings** | | | | |
| Yes | 47 (82.5) | 10 (17.5) | 8.225 | 0.004* |
| No | 91 (61.5) | 57 (38.5) | | |

*Significance at p<0.05, ** Fisher's exact test.

times (aOR=3.206, 95% CI = 91.66-6.17), more likely to practice SRHR than those women who did not discuss. Similarly, women having adequate knowledge on SRHR were 3.234 times (aOR=3.234, 95% CI = 1.85-6.56) more likely to practice than those who had inadequate knowledge (Table 6).

## Discussion

This study revealed that married women of reproductive age had inadequate knowledge and poor practice of SRHR, and only 47.7% of the women had adequate knowledge. This is lower compared to the proportion among Madhesi women of reproductive age in Sarlahi district [16], probably due to the differences in the study population and site. Similarly, in Kapan VDC, Kathmandu, there was also a high level of knowledge on reproductive rights probably reflecting better access to information in the capital city [6]. On the contrary, the studies conducted by Makinde OA, Adebayo AM and Tadesse T et al. showed less than half of the participants had enough knowledge on SRHR, which corresponds to the result of the present study [20,21]. A study conducted among undergraduate students in Tanahun, Nepal, however recorded a higher level of knowledge probably because students are more exposed to educational resources regarding SRHR compared to married women [22]. The study by Ogunlayi MA conducted in southwestern Nigeria reported that 32% of in-school and 40.5% of out-of-school adolescents recognized the right to privacy and confidentiality as part of SRHR, higher than this study [23]. This might be due to the difference in the study population. Regarding age-related differences, this study found that women aged 18–33 years were more likely to have adequate SRHR knowledge compared to the age group of 34–49 years. This contrasts with findings by Hossain MD et al., where SRHR knowledge increased with advancing age, possibly due to sociocultural and contextual differences [24]. Furthermore, less than half of the participants in Tadesse T et al.'s study knew the minimum legal age of marriage, which is lower than in this study, potentially due to variations in socio-economic status, education, and access to SRHR information [20].

The knowledge of SRHR was significantly related to the current economic status. Corresponding to this finding, the cross-sectional study conducted in Lahore, Pakistan, also revealed that the women from the better economic group showed higher knowledge about SRHR than the below the poverty line category [25]. Gebretsadik GG and Weldearegay reported that the higher the socioeconomic status, the better the awareness about sexual and reproductive health due to wider access to information and resources [26]. Conversely, women of lower economic status may face financial barriers to accessing SRHR information. In the current study, majority of the participants reported the correct age of pregnancy, which is comparable to studies conducted among similar groups in Nepal and India [19,27,28]. Regarding sources of information, majority of the respondents in the present study reported mass media to be their main source of information on SRHR, which is supported by findings by Yadav RK et al., Ashebir W et al., and Gebretsadik GG and Weldearegay GG [15,26,29]. This suggests that media such as radio and TV play a crucial role in delivering reproductive health information in developing nations because most of the time, these channels are accessible free of cost. In contrast, a majority of respondents reportedly relied on radio/TV as a source of reproductive health information in a study by Adinew YM et al., likely due to the greater accessibility of these media channels among their population [19]. Three-fourths respondents said

**Table 4. Multiple logistic regression between independent variables and level of knowledge.**

| Variables | cOR | 95% CI | p-value | aOR | 95% CI | p-value |
|---|---|---|---|---|---|---|
| **Age** | | | | | | |
| 18-33 | 1.860 | 1.21-2.86 | 0.005 | 1.811 | 1.14-2.86 | 0.011* |
| 34-49 | Ref. | | | Ref. | | |
| **Age at first marriage** | | | | | | |
| < 18 years | Ref. | | | Ref. | | |
| ≥18 years | 3.069 | 1.79-5.26 | <0.001 | 2.075 | 1.16-3.69 | 0.014* |
| **Occupation** | | | | | | |
| Formal sectors | 2.142 | 1.38-3.30 | 0.001 | 1.834 | 1.15-2.90 | 0.010* |
| Informal sectors | Ref. | | | Ref. | | |
| **Economic status** | | | | | | |
| Below the poverty line | Ref. | | | Ref. | | |
| Above the poverty line | 3.111 | 1.95-4.96 | <0.001 | 2.511 | 1.52-4.14 | <0.001* |
| **Educational level of respondent** | | | | | | |
| Illiterate | Ref. | | | Ref. | | |
| Literate only | 6.470 | 2.24-18.7 | <0.001 | 3.352 | 1.43-15.9 | 0.04* |
| Primary education | 15.800 | 5.46-45.9 | <0.001 | 9.871 | 2.67-21.2 | <0.001* |
| Secondary education | 5.480 | 1.95-15.4 | 0.001 | 6.670 | 1.76-11.2 | 0.01* |
| Higher education | 4.370 | 1.48-13.0 | 0.007 | 7.367 | 2.34-13.4 | 0.03* |
| **Educational level of husband** | | | | | | |
| Illiterate | Ref. | | | Ref. | | |
| Literate only | 0.600 | 0.05-6.21 | 0.66 | 0.711 | 0.30-1.06 | 0.87 |
| Primary education | 0.472 | 0.05-4.19 | 0.50 | 0.872 | 0.43-0.97 | 0.67 |
| Secondary education | 0.172 | 0.02-1.46 | 0.10 | 0.357 | 0.11-0.86 | 0.45 |
| Higher education | 0.081 | 0.009-0.6 | 0.02 | 1.872 | 1.20-4.87 | 0.02* |
| **Discussion of SRH issues with others** | | | | | | |
| Yes | 0.756 | 0.38-1.48 | 0.001 | 0.321 | 0.22-0.73 | 0.23 |
| No | Ref. | | | Ref. | | |
| **Source of information from awareness programs and trainings** | | | | | | |
| Yes | 1.340 | 1.15-3.72 | 0.004 | 2.134 | 1.72-5.67 | 0.021* |
| No | Ref. | | | Ref. | | |

*Significance at p<0.05.

STIs are the consequences of unsafe sex, which is lower than the study conducted by Tafa Segni M et al., where most of the respondents responded that STIs are major consequences of unsafe sex [17]. This might be due to differences in study populations as students might be more familiar with unsafe sex and its consequences through textbooks.

In this study, more than half of the respondents reported poor practice about SRHR. This depicts the poor decision-making power of women in reproductive health, less discussion of SRH issues, and low involvement of respondents in RH clubs/mother groups. One-fourth of the respondents reported early-age marriage according to the findings of this study. Despite the law against early marriage in the country, still early marriage is still practiced due to the social, cultural, religious, and educational systems of our country, and most of the women are compelled to do early marriage. Early marriage can also mark the end of a girl's education, hence depriving her of the associated benefits. Young married girls are also more likely to be exposed to domestic violence and sexual abuse. They also face a high risk of forced sex and HIV [27]. In this study majority of the respondents made combine decision by both wives and husbands in regards

**Table 5. Bivariate analysis of independent variables with practice of SRHR.**

| Variables | Practice of SRHR | | χ 2 | p-value |
|---|---|---|---|---|
| | Good n (%) | Poor n (%) | | |
| **Age** | | | | |
| 18-33 | 74 (42) | 102 (58) | 0.041 | 0.839 |
| 34-49 | 68 (41) | 98 (59) | | |
| **Ethnicity** | | | | |
| Brahmin/ Chhetri | 78 (47.3) | 87 (52.7) | 5.831 | 0.064 |
| Janajati | 44 (39.6) | 67 (60.4) | | |
| Others | 20 (30.3) | 46 (69.7) | | |
| **Religion** | | | | |
| Hindu | 33 (34) | 64 (66) | 3.137 | 0.077 |
| Non-Hindu | 109 (44.5) | 123 (50.2) | | |
| **Family type** | | | | |
| Nuclear family | 77 (47) | 87 (53) | 3.827 | 0.060 |
| Others | 65 (36.5) | 113 (63.5) | | |
| **Type of marriage** | | | | |
| Love marriage | 33 (31.1) | 73 (68.9) | 6.827 | 0.009* |
| Arrange marriage | 109 (46.2) | 127 (53.8) | | |
| **Educational level of respondent** | | | | |
| Illiterate | 17 (29.8) | 40 (70.2) | 5.71 | 0.22 |
| Literate only | 31 (44.9) | 38 (55.1) | | |
| Primary education | 34 (38.6) | 54 (61.4) | | |
| Secondary education | 46 (48.4) | 49 (51.6) | | |
| Higher education | 14 (42.4) | 19 (57.6) | | |
| **Husband's educational level** | | | | |
| Literate only | 7 (23.3) | 23 (76.7) | 12.30 | 0.006* |
| Primary education | 20 (29) | 49 (71) | | |
| Secondary education | 62 (45.6) | 74 (54.4) | | |
| Higher education | 53 (49.5) | 54 (50.5) | | |
| **Age at first marriage** | | | | |
| < 18 years | 22 (26.5) | 61 (73.5) | 10.176 | 0.001* |
| ≥18 years | 120 (46.3) | 139 (53.7) | | |
| **Occupational status** | | | | |
| Formal sectors | 89 (47.6) | 98 (43.9) | 6.267 | 0.012* |
| Informal sectors | 53 (34.2) | 102 (65.8) | | |
| **Husband's occupational status** | | | | |
| Formal sectors | 134 (42.7) | 180 (57.3) | 2.106 | 0.147 |
| Informal sectors | 8 (28.6) | 20 (71.4) | | |
| **Economic status** | | | | |
| Below poverty line | 36 (28.8) | 89 (71.2) | 13.129 | <0.001* |
| Above poverty line | 106 (48.8) | 111 (51.2) | | |
| **Participation at health clubs/mothers group** | | | | |
| Participated | 75 (60.5) | 49 (39.5) | 28.811 | <0.001* |
| Not participated | 67 (30.7) | 151 (69.5) | | |
| **Discussion of SRH issues with others** | | | | |
| Yes | 161 (50) | 161 (50) | 12.078 | 0.001* |

*(Continued)*

**Table 5.** (Continued)

| Variables | Practice of SRHR | | χ 2 | p-value |
|---|---|---|---|---|
| | Good n (%) | Poor n (%) | | |
| No | 2 (10) | 18 (90) | | |
| **Source of information from awareness programs/trainings** | | | | |
| Yes | 42 (73.7) | 15 (26.3) | 4.611 | 0.032* |
| No | 85 (57.4) | 63 (42.6) | | |

*Significance at p<0.05.

to giving birth to a child which is higher than the study conducted in another districts of Nepal [6,15]. This could be due to more involvement of the husband in making decisions relating to SRH. In the present study, a few of the respondents made decisions in regard to birth spacing between the children. Most of the women had made the decision together with the husband, and in rare events, by family members and health workers. The finding of the study is similar to the finding of a similar study conducted in Bolivia, which showed that the decision regarding family planning was mostly taken by couples, followed by women only and by men only [30]. Women's participation in joint decision-making may be taken as encouraging. Yet, husbands' domination is evident. In the study, less than half were involved in RH clubs/mothers groups which is similar to the mixed-method study conducted among marginalized women in Nepal [18]. Respondents who had participated in reproductive health clubs/mothers groups were found to be significantly associated with practice on SRHR, which is similar to the study conducted in Ethiopia [26,29]. This is because such clubs and groups can provide an opportunity to ask and discuss SRH issues and rights. In this study, the participants who had at least an ANC checkup and a PNC checkup and had an institution delivery was higher than the findings of the annual report [31].

The current study has a number of strengths. It pertains to a current and important health issue via the study of knowledge and practices of SRHR. The analyses of socio-demographic factors associated with knowledge and utilization of SRHR identified main elements that must be addressed in order to enhance uptake. In this regard, the study puts forward practical education, improvement of economic status, and late marriage as actionable recommendations for future interventions and policy development. This study also has a few limitations. The study was limited to a semi-urban setting; such a setting may not be very representative of the knowledge and practices in rural settings, where the knowledge and utilization of SRHR might be drastically different. Further, reliance upon self-reported information brings in a number of biases, notably social desirability bias and recall bias, that could affect the accuracy of results. The absence of qualitative research methods further limits the in-depth understanding of barriers and facilitators that surround the context-specific knowledge and use of SRHR. These are important limitations that raise avenues for further research on sexual and reproductive health and rights.

## Conclusion

Adequate knowledge and understanding of SRHR are critical to the ability to protect women from violation of those rights. This study had assessed the knowledge and practice of sexual and reproductive health rights among the married women of reproductive age, where the majority of the respondents neither had adequate knowledge nor had good practice of SRHR. Less than half of the respondents had adequate knowledge and good practice of SRHR. A large proportion of married women of reproductive age were lacking knowledge about SRHR and were not able to practice their rights related to sexual and reproductive health. It is essential to make women aware of their rights and empower them to the level where they can utilize their rights. For that, awareness programs relating to SRHR should be designed, focusing on the reproductive age groups, which enables women to know and exercise their rights effectively in the future. Since there is

**Table 6. Multiple logistic regression between independent variables and level of practice.**

| Variables | cOR | 95% CI | p-value | aOR | 95% CI | p-value |
|---|---|---|---|---|---|---|
| **Marriage type** | | | | | | |
| Love marriage | Ref. | | | Ref. | | |
| Arrange marriage | 1.899 | 1.17-3.08 | 0.009 | 1.768 | 1.07-2.91 | 0.025* |
| **Age at first marriage** | | | | | | |
| < 18 years | Ref. | | | Ref. | | |
| ≥18 years | 2.394 | 1.38-4.12 | 0.002 | 1.821 | 1.02-3.24 | 0.042* |
| **Occupation** | | | | | | |
| Formal sectors | 1.748 | 1.12-2.71 | 0.013 | 1.518 | 0.96-2.39 | 0.072 |
| Informal sectors | Ref. | | | Ref. | | |
| **Economic status** | | | | | | |
| Below the poverty line | Ref. | | | Ref. | | |
| Above the poverty line | 2.361 | 1.47-3.77 | <0.001 | 1.900 | 1.15-3.12 | 0.012* |
| **Husbands' educational level** | | | | | | |
| Literate only | Ref. | | | Ref. | | |
| Primary education | 0.746 | 0.27-2.01 | 0.56 | 0.453 | 0.21-5.93 | 0.74 |
| Secondary education | 0.363 | 0.36-0.14 | 0.03 | 0.734 | 0.49-0.95 | 0.06 |
| Higher education | 0.310 | 0.12-0.78 | 0.01 | 0.345 | 0.15-0.87 | 0.04* |
| **Participation at health clubs/mothers group** | | | | | | |
| Participated | 3.450 | 2.18-5.47 | <0.001 | 4.213 | 2.35-7.69 | <0.001* |
| Not participated | Ref. | | | Ref. | | |
| **Discussion of SRH issues with others** | | | | | | |
| Yes | 2.364 | 1.63-4.82 | 0.001 | 3.206 | 1.66-6.17 | <0.001* |
| No | Ref. | | | Ref. | | |
| **Source of information from awareness programs/trainings** | | | | | | |
| Yes | 0.482 | 0.246-0.94 | 0.03 | 1.606 | 0.780-3.306 | 0.198 |
| No | Ref. | | | Ref. | | |
| **Knowledge** | | | | | | |
| Adequate | 2.720 | 1.74-4.23 | <0.001 | 3.234 | 1.85-6.56 | <0.001* |
| Inadequate | Ref. | | | Ref. | | |

*Significance at $p < 0.05$.

limited study on this topic, further research could be done, which will help to focus on other factors that have an impact on knowledge and practice of SRHR.

## Supporting information

**S1 Text. Sampling Procedure.**
(DOCX)

**S2 Text. Research Tools.**
(DOCX)

**S1 Data. Data.**
(SAV)

## Acknowledgments

The authors would like to acknowledge the support of all the members of the faculty of public health at the Manmohan Memorial Institute of Health Sciences, Authorities of Besishahar Municipality and special thanks to all the participants of this study.

## Author contributions

**Conceptualization:** Laxmi Gautam, Srijana Adhikari, Amrit Bist.

**Data curation:** Srijana Adhikari.

**Formal analysis:** Laxmi Gautam, Srijana Adhikari, Amrit Bist.

**Funding acquisition:** Laxmi Gautam, Srijana Adhikari.

**Investigation:** Srijana Adhikari.

**Methodology:** Laxmi Gautam, Srijana Adhikari, Amrit Bist, Sujan Gautam.

**Project administration:** Laxmi Gautam, Srijana Adhikari.

**Resources:** Laxmi Gautam, Srijana Adhikari.

**Software:** Laxmi Gautam, Srijana Adhikari, Amrit Bist, Sujan Gautam.

**Supervision:** Laxmi Gautam, Sujan Gautam.

**Validation:** Laxmi Gautam, Sujan Gautam.

**Visualization:** Laxmi Gautam.

**Writing – original draft:** Laxmi Gautam, Amrit Bist.

**Writing – review & editing:** Laxmi Gautam, Amrit Bist, Sujan Gautam.

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
