## [Decision Letter · Decision Letter 0]

29 May 2025

PGPH-D-25-00425

Sexual and reproductive health rights: knowledge and practice among the married women of reproductive age residing in Besishahar Municipality, Nepal

Dear Dr. Gautam,

Thank you for submitting your manuscript to PLOS Global Public Health. After careful consideration, we feel that it has merit but does not fully meet PLOS Global Public Health’s publication criteria as it currently stands. Therefore, we invite you to submit a revised version of the manuscript that addresses the points raised during the review process.

Two reviewers have provided their feedback on your manuscript. Please see their comments below and in the attached file. We ask that you carefully consider their suggestions and revise your manuscript accordingly, providing a point-by-point response to the reviewers.

We look forward to receiving your revised manuscript.

Kind regards,

Sarah Jose, Ph.D.

Staff Editor

Journal Requirements:

1. Please provide additional details regarding participant consent. In the ethics statement in the Methods and online submission information, please ensure that you have specified (1) whether consent was informed and (2) what type you obtained (for instance, written or verbal, and if verbal, how it was documented and witnessed).

2. In the online submission form, you indicated that "Data will be provided by the corresponding author in a reasonable request".

a. In a public repository,

b. Within the manuscript itself, or

c. Uploaded as supplementary information.

Additional Editor Comments (if provided):

Reviewers' comments:

Reviewer's Responses to Questions

**Comments to the Author**

1. Does this manuscript meet PLOS Global Public Health’s publication criteria? Is the manuscript technically sound, and do the data support the conclusions? The manuscript must describe methodologically and ethically rigorous research with conclusions that are appropriately drawn based on the data presented.

Reviewer #1: Yes

Reviewer #2: Yes

2. Has the statistical analysis been performed appropriately and rigorously?

Reviewer #1: Yes

Reviewer #2: No

3. Have the authors made all data underlying the findings in their manuscript fully available (please refer to the Data Availability Statement at the start of the manuscript PDF file)?

Reviewer #1: No

Reviewer #2: Yes

4. Is the manuscript presented in an intelligible fashion and written in standard English?

Reviewer #1: Yes

Reviewer #2: Yes

5. Review Comments to the Author

Reviewer #1: This is a relevant study. Several areas require clarification and refinement

Title: Consider revising the title to explicitly mention the study design, e.g. “A Cross-Sectional Study of…”

Abstract:

The statement “women married at an appropriate age were 2.075 times more likely…” should specify what is meant by “appropriate age.”

The second part of the sentence referring to “which was 1.82 (aOR=1.821, 95% CI=1.02–3.24) in the case of good practice” lacks clarity. Consider rephrasing for better readability and separation of findings on knowledge and practice.

Introduction:

Line 69: Please include a citation to support the 24.6% unmet need for family planning.

Line 71: Similarly, provide a citation for the claim regarding unsafe abortions.

Methods:

Line 142: The authors mention that the study was based on STROBE. Since this is a cross-sectional study, it would be useful to elaborate on how the STROBE guidelines were applied or used to structure the study/reporting.

The process of scoring for SRHR knowledge and practices is not clearly described. A detailed explanation is needed in the methods section to support the interpretation of Table 2.

Results

The choice of age groupings (18-33) should be justified. Why not use WHO-standard classifications? Moreover, age ranges in the bivariate analysis table differ from those in the sociodemographic table (Table 1). Please ensure consistency or explain the rationale for the variations.

Clarify what is meant by “literate only” under the education variable. Does this refer to individuals who can read/write but have no formal education?

The categorization of occupation (services, business, private sector) is ambiguous. Consider merging or redefining these categories for clarity.

How was the poverty line determined in this study? Was a standardized tool such as the Poverty Probability Index (PPI) used? Please describe the criteria or methodology used.

Reviewer #2: Sexual and Reproductive health Rights: knowledge and practice among the married women of reproductive age residing in Besishahar Municipality, Nepal is well addressed in this study. Please consider the comments in the attached file.

6. PLOS authors have the option to publish the peer review history of their article (what does this mean?). If published, this will include your full peer review and any attached files.

**Do you want your identity to be public for this peer review?** For information about this choice, including consent withdrawal, please see our Privacy Policy.

Reviewer #1: No

Reviewer #2: **Yes: **Soghra Khani

---

## [Editor Report · Decision Letter 1]

12 Sep 2025

Sexual and reproductive health rights: A cross-sectional study of knowledge and practice among the married women of reproductive age residing in Besishahar Municipality, Nepal

PGPH-D-25-00425R1

Dear Ms Gautam,

We are pleased to inform you that your manuscript 'Sexual and reproductive health rights: A cross-sectional study of knowledge and practice among the married women of reproductive age residing in Besishahar Municipality, Nepal' has been provisionally accepted for publication in PLOS Global Public Health.

Best regards,

Dr Tanmay Bagade, Ph.D., MS (O&G), MPH, MHM

Academic Editor